# Asymmetric construction of tetrahedral chiral zinc with high configurational stability and catalytic activity

Kenichi Endo [1,2], Yuanfei Liu [1], Hitoshi Ube [1], Koichi Nagata [1,3] & Mitsuhiko Shionoya [1✉]

Chiral metal complexes show promise as asymmetric catalysts and optical materials. Chiral-at-metal complexes composed of achiral ligands have expanded the versatility and applicability of chiral metal complexes, especially for octahedral and half-sandwich complexes. However, Werner-type tetrahedral complexes with a stereogenic metal centre are rarely used as chiral-at-metal complexes because they are too labile to ensure the absolute configuration of the metal centre. Here we report the asymmetric construction of a tetrahedral chiral-at-zinc complex with high configurational stability, using an unsymmetric tridentate ligand. Coordination/substitution of a chiral auxiliary ligand on zinc followed by crystallisation yields an enantiopure chiral-only-at-zinc complex (> 99% ee). The enantiomer excess remains very high at 99% ee even after heating at 70 °C in benzene for one week. With this configurationally stable zinc complex of the tridentate ligand, the remaining one labile site on the zinc can be used for a highly selective asymmetric oxa-Diels-Alder reaction (98% yield, 87% ee) without substantial racemisation.

[1] Department of Chemistry, Graduate School of Science, The University of Tokyo, 7-3-1 Hongo, Bunkyo-ku, Tokyo 113-0033, Japan. [2] Present address: Institute for Chemical Research, Kyoto University, Gokasho, Uji, Kyoto 611-0011, Japan. [3] Present address: Department of Chemistry, Graduate School of Science, Tohoku University, Aoba-ku, Sendai, Miyagi 980-8578, Japan. ✉email: shionoya@chem.s.u-tokyo.ac.jp

Enantiopure chiral metal complexes are widely used for catalytic[1], chiroptical[2], medical[3] and molecular recognition[4] purposes. Common synthetic methods use enantiopure chiral ligands[5–7] or counterions[8] as components. An alternative is to coordinate ligands to a metal atom in a chiral configuration[9], which is known as chiral-at-metal complexes[10]. The latter method is recently gathering attention because the central metal atom can act as a stereocentre as well as a substrate activation centre without other chiral constituents[11,12]. However, previous examples of enantiopure complexes with metal-centred chirality have focused on octahedral[13–22] and half-sandwich pseudotetrahedral[23–25] coordination geometries. Although Werner-type tetrahedral metal complexes can have structures similar to carbon centres to which four different substituents can bind, their stereoinversion is generally very fast, making it difficult to maintain long-term enantiopurity[26] (Fig. 1a). It has therefore long been a difficult task to construct and utilise Werner-type tetrahedral chiral metal centres without using chiral constituents.

Zinc shows promise as a metal ion capable of forming a tetrahedral coordination structure, as exemplified by zinc enzymes in biological systems. For example, in the active centres of zinc enzymes such as carbonic anhydrase and alcohol dehydrogenase, the flexible coordination structures and Lewis acidity unique to zinc complexes are successfully used[27]. In these enzymes, hydration and reduction reactions, respectively, occur in the relatively labile fourth coordination site in the chiral catalytic centre where three different amino acid residues are coordinated to the zinc centre. Focusing on the stable chiral structures and functions of such zinc enzymes, we set out to construct a stable zinc-centred chirality using achiral constituents that enables asymmetric catalysis.

In this study, we establish a highly enantioselective synthesis of tetrahedral zinc complexes with long-term stability using an unsymmetric achiral tridentate ligand (Fig. 1b). We also develop a highly selective asymmetric catalytic reaction by utilising the relatively labile coordination site of the obtained optically pure zinc complex and clarify the reaction mechanism.

## Results

**Ligand design**. We have designed an unsymmetric, achiral tridentate ligand for a tetrahedral chiral-at-zinc complex. The fourth coordination site of the zinc complex was postulated as a substrate binding site for asymmetric induction and catalysis. One problem to be solved is the fast inversion of the absolute configuration of the zinc centre, which leads to racemisation. There are two possible mechanisms for this stereoinversion. Firstly, stereoinversion can occur through complete or partial dissociation of the tridentate ligand from the zinc centre followed by reassociation without asymmetric induction (Fig. 2a). Another possible mechanism is the stereoinversion between the two tetrahedral enantiomers through some states in which the tridentate ligand does not dissociate and is flush with the zinc centre (Fig. 2b). In other words, to prevent racemisation of the zinc centre, it is necessary to pay attention to the structure of the ligand and the properties of the solvent to suppress the flattening of the zinc complex structure and the dissociation of the tridentate ligand.

From this perspective, an achiral unsymmetrical tridentate ligand $H_2L$ was designed and synthesised for tetrahedral chiral-at-zinc complexes, [ZnLL$^4$] (L$^4$ = an achiral monodentate ligand) (Fig. 1b). The ligand $L^{2-}$ after deprotonation has a dianionic charge and two rigid chelate rings, and suppresses dissociation of the tridentate ligand that promotes racemisation. Even if one end of $L^{2-}$ dissociates, the three donor atoms cannot be in the same plane as the zinc atom due to the steric hindrance and strain, thus preventing stereoinversion caused by partial dissociation or flattening (Supplementary Fig. 1). These two structural factors were expected to allow $L^{2-}$ to maintain and stabilise the absolute configuration of the tetrahedral zinc centre, even in the presence of a labile coordination site.

**Asymmetric construction of a tetrahedral chiral zinc centre**. To obtain enantiopure chiral substances, enantioselective or diastereoselective synthesis is an effective tool alongside optical resolution techniques such as crystallisation or chromatography.

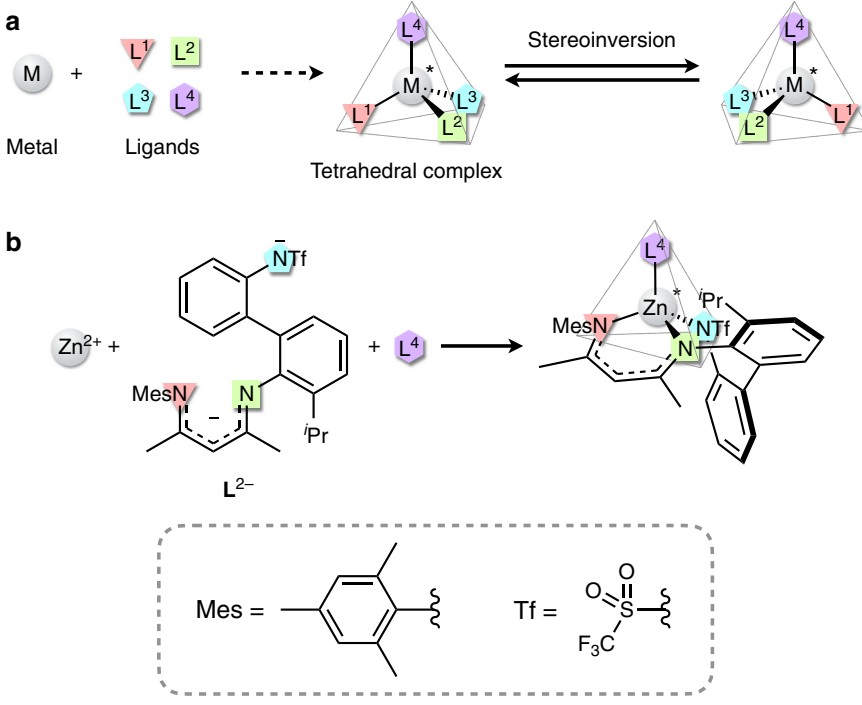

**Fig. 1 Design for a chiral tetrahedral metal complex with a configurationally stable metal centre. a** Construction of a chiral tetrahedral metal complex using four different monodentate ligands and stereoinversion between its enantiomers. **b** Construction of a chiral tetrahedral zinc complex with $L^{2-}$.

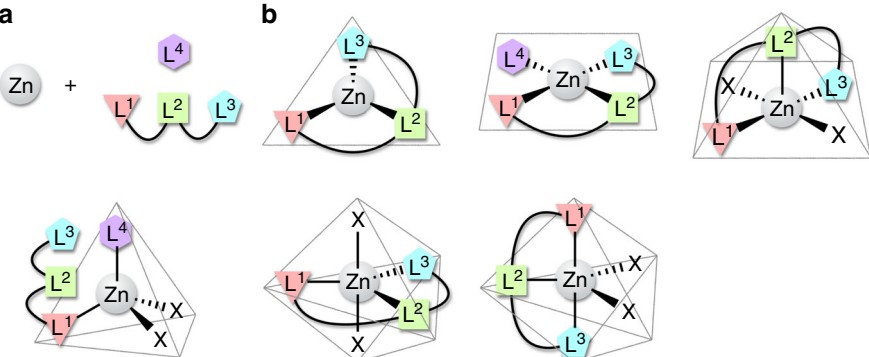

**Fig. 2 Possible stereoinversion pathways of a tetrahedral zinc centre coordinated by an achiral, unsymmetric tridentate ligand. a** Stereoinversion pathways of stereoinversion by complete or partial dissociation of the tridentate ligand. **b** Some stereoinversion pathways without dissociating the tridentate ligand. X arbitrary monodentate ligand.

A few excellent examples of asymmetric synthesis of enantiopure chiral octahedral metal complexes have been reported so far using chiral auxiliary ligands[28]. Inspired by these reports, we developed an asymmetric synthesis of an enantiopure chiral tetrahedral zinc complex using an unsymmetric achiral tridentate ligand $L^{2-}$ and a chiral auxiliary in the following three steps: (i) complexation of tridentate $L^{2-}$ and a source of zinc to prepare a racemic mixture of tetrahedral zinc complex; (ii) asymmetric induction at the zinc centre using an appropriate coordinating chiral auxiliary; (iii) replacement of the chiral auxiliary ligand with an achiral ligand while maintaining the absolute configuration of the zinc centre.

Firstly, ligand $H_2L$ was reacted with diethylzinc to obtain a racemic mixture of dimeric complexes, $rac$-[$Zn_2L_2$] (Fig. 3a). The crystal includes ($R_{Zn}$,$R_{Zn}$)-[$Zn_2L_2$] and ($S_{Zn}$,$S_{Zn}$)-[$Zn_2L_2$] in a 1:1 ratio, as was determined by single-crystal X-ray structure analysis ($R_{Zn}$ and $S_{Zn}$ denote the absolute configuration of each zinc centre). Figure 3b shows only the portion ($R_{Zn}$,$R_{Zn}$)-[$Zn_2L_2$]. In this structure, the zinc centre is bound by the doubly deprotonated tridentate ligand $L^{2-}$ and the sulfonyl oxygen donor of the other $L^{2-}$, forming a dimer of the four-coordinate zinc complex. That the tetrahedral structure is maintained in solution was supported by $^1H$ nuclear magnetic resonance (NMR) measurements including NOESY in $C_6D_6$ (Supplementary Fig. 2). As shown below, this dimer is believed to dissociate into monomers by ligand exchange with an appropriate monodentate ligand.

Secondly, $rac$-[$Zn_2L_2$] dimer was reacted with a chiral auxiliary, ($S$)-dpp (dpp = α,α-diphenyl-2-pyrrolidinemethanol) (2.5 equiv.), in aprotic $C_6D_6$ as the fourth ligand to control the absolute configuration of the zinc centre (Fig. 3a). At the initial stage of this reaction, two diastereomers, ($S_{Zn}$)-[$ZnL$(($S$)-dpp)] and ($R_{Zn}$)-[$ZnL$(($S$)-dpp)], were generated in a ratio of about 1:1, as demonstrated by $^1H$ and $^{19}F$ NMR spectroscopy (Fig. 4a, b). Upon heating at 70 °C for 48 h and standing at room temperature for 24 h, the diastereomer ratio gradually changed and finally reached 51:1.

This equilibrium shift involves configurational changes at the zinc centre, although the ligand $L^{2-}$ was designed to slow the configurational isomerisation. To examine the effect of the hydroxy group of ($S$)-dpp on the stereoinversion, the isomerisation kinetics was compared with ($S$)-mdp (mdp = 2-(methoxydiphenylmethyl)pyrrolidine) where the hydroxyl group of ($S$)-dpp is methylated (Fig. 4c). As a result, it was found that ($S$)-dpp bearing a hydroxy group significantly accelerated the stereoinversion (Fig. 4b). Therefore, the ($S$)-dpp acts not only as a chiral auxiliary, but also as a promoter of stereoinversion at the zinc centre. The stereoinversion was further accelerated by adding an excess of ($S$)-dpp (Fig. 4b). As shown below, when the ($S$)-dpp is

replaced by an achiral monodentate ligand and the dissociated ($S$)-dpp is removed, the absolute configuration of the zinc centre was retained and kept stable for a long time.

The structures of ($S_{Zn}$)-[$ZnL$(($S$)-dpp)] in the crystal state and in solution were determined by single-crystal X-ray diffraction (Fig. 3b) and $^1H$ NMR spectroscopy including NOESY (Supplementary Fig. 2), respectively. In the crystal structure, two NH–O and one OH–π hydrogen bonds were found (Fig. 3b and Supplementary Fig. 3). The equilibrium in solution between the diastereomers was biased toward the ($S_{Zn}$) form, most likely because of the hydrogen bonding and steric repulsion observed between ($S$)-dpp and $L^{2-}$.

Thirdly, the diastereo-enriched [$ZnL$(($S$)-dpp)] (($S_{Zn}$)-[$ZnL$(($S$)-dpp)]:($R_{Zn}$)-[$ZnL$(($S$)-dpp)] = 51:1) was reacted with $^tBuCN$ to replace ($S$)-dpp by achiral $^tBuCN$ (Fig. 3a). The product ($S_{Zn}$)-[$ZnL$(NC$^tBu$)] was isolated by crystallisation in 72% yield (three steps from $H_2L$). The crystal structure of [$ZnL$(NC$^tBu$)] was confirmed by single-crystal X-ray diffraction (Fig. 3b), and was found to be maintained in solution as shown by $^1H$ NMR spectroscopy including NOESY (Supplementary Fig. 2). The $S_{Zn}$ configuration at the zinc centre was confirmed by the anomalous dispersion in single-crystal X-ray diffraction.

The enantiomeric excess (ee) of the crystalline ($S_{Zn}$)-[$ZnL$(NC$^tBu$)] was determined in $C_6D_6$ to be > 99% ee by $^{19}F$ NMR spectroscopy using a chiral shift reagent, ($R$)-mts (mts = methyl $p$-tolyl sulfoxide) (Fig. 5a, b). First, we confirmed that the addition of ($R$)-mts to $rac$-[$Zn_2L_2$] yields two completely separated signals with a 1:1 integration ratio (Fig. 5a). This experiment validated the accuracy of this method for determining the optical purity of the product. The optical purity of the obtained ($S_{Zn}$)-[$ZnL$(NC$^tBu$)] was confirmed by the same method, and it was found that the enantiomeric excess was 99% ee or more (Fig. 5b).

The product ($S_{Zn}$)-[$ZnL$(NC$^tBu$)] has a chiral structure based on a tetrahedral zinc centre. The geometry of the zinc centre of the crystal structure was almost tetrahedral, as shown by the geometry index[29] $\tau_4 = 0.85$. The ligands $L^{2-}$ and $^tBuCN$ are achiral, but four chemically inequivalent nitrogen atoms coordinate to the zinc centre, forming a tetrahedral chiral-at-zinc structure. The Cotton effects of ($S_{Zn}$)-[$ZnL$(NC$^tBu$)] were also confirmed in its CD spectrum (Supplementary Fig. 4). The complex ($R_{Zn}$)-[$ZnL$(NC$^tBu$)] synthesised by the same method using ($R$)-dpp showed the expected mirror-image CD spectrum from that of ($S_{Zn}$)-[$ZnL$(NC$^tBu$)]. In this way, by combining the designed tridentate ligand $L^{2-}$ and the chiral auxiliary ligand ($S$)-dpp, a tetrahedral chiral zinc with a single configuration can be constructed with high enantioselectivity.

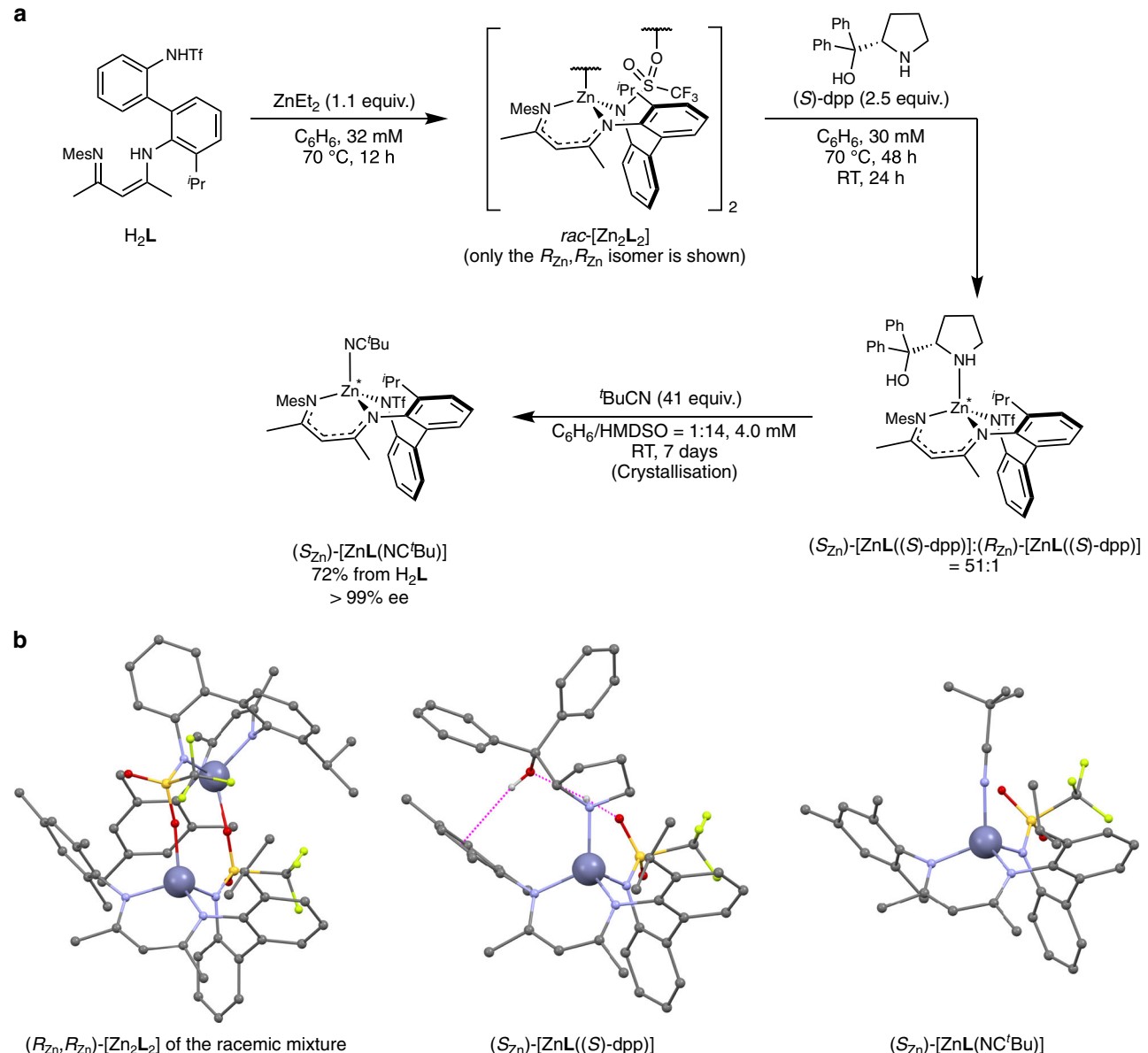

**Fig. 3 Synthesis of enantiopure ($S_{Zn}$)-[ZnL(NC$^t$Bu)] by highly diastereoselective asymmetric induction using a coordinating chiral auxiliary followed by replacement with an achiral $^t$BuCN molecule. a** Reaction scheme. RT room temperature, HMDSO hexamethyldisiloxane, equiv. equivalents, ee enantiomer excess. **b** X-ray single-crystal structures of three zinc complexes in a ball-and-stick model. Hydrogen atoms on carbon atoms and solvent molecules are omitted for clarity. For ($S_{Zn}$)-[ZnL((S)-dpp)], one of the crystallographically independent but structurally similar molecules is shown. Colour code: Zn, blue grey; C, grey; N, blue; O, red; F, yellow green; S, yellow. Dashed magenta lines denote plausible hydrogen bonds.

**Configurational stability of the tetrahedral chiral zinc centre.**
Next, we investigated the configurational stability of the zinc centre using a physically and enantiomerically pure complex, ($S_{Zn}$)-[ZnL(NC$^t$Bu)]. Even when ($S_{Zn}$)-[ZnL(NC$^t$Bu)] was heated in aprotic C$_6$D$_6$ in a dry N$_2$ atmosphere at 70 °C for one week, the enantiomer excess was maintained as highly as 99% ee (Fig. 5c). This very high stability is in stark contrast to the typical tetrahedral zinc complexes which have been reported to undergo an instantaneous stereoinversion[30,31]. This result demonstrates the effectiveness of the design of L$^{2-}$ in slowing down stereoinversion, even when the tetrahedral four-coordinate structure has a labile coordination site.

The effect of different solvents on the configurational stability of the zinc centre was also investigated (Supplementary Table 1). For example, in protic $^i$PrOH the stereoinversion was accelerated whereas in aprotic polar solvents (THF, CH$_2$Cl$_2$ and CH$_3$CN), the stereoinversion was only slightly accelerated. The results indicate

that ($S_{Zn}$)-[ZnL(NC$^t$Bu)] can be used in various aprotic solvents with little loss of enantiopurity.

**Catalytic activity of the tetrahedral chiral zinc centre.** Finally, to clarify the reactivity of the labile coordination site of ($S_{Zn}$)-[ZnL(NC$^t$Bu)], we investigated its catalytic function in asymmetric reactions. The Lewis acidity of this chiral zinc complex was effectively utilised in the asymmetric oxa-Diels–Alder catalytic reaction[32,33]. Specifically, a reaction of Danishefsky's diene **1** with aldehyde **2** was performed in the presence of 2 mol% of ($S_{Zn}$)-[ZnL(NC$^t$Bu)] (Fig. 6a) at 19 °C for 24 h. After acidic workup, adduct (R)-**3** was obtained in 98% yield with 87% ee (Supplementary Fig. 5). However, compound **3** was not produced at all when the reaction was carried out without the zinc catalyst, indicating that the complex ($S_{Zn}$)-[ZnL(NC$^t$Bu)] acts as a chiral Lewis acid catalyst.

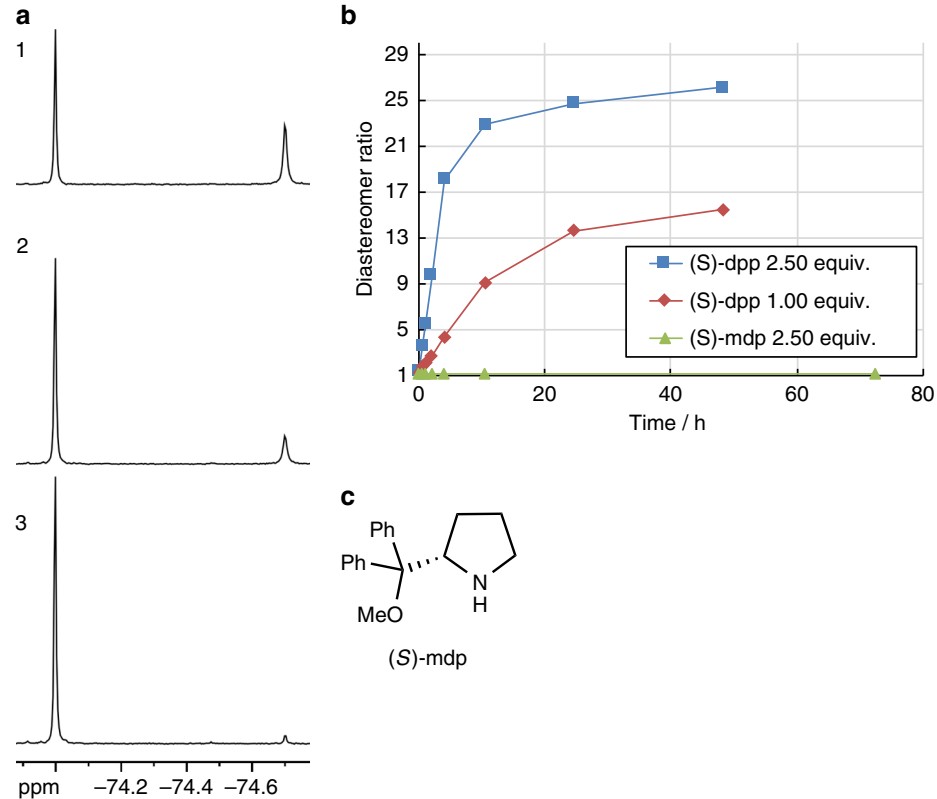

**Fig. 4 Process of the dynamic asymmetric induction in [ZnL((S)-dpp)]. a** $^{19}$F NMR spectra (C$_6$D$_6$, 300 K, 471 MHz) for the time-course analysis for the formation of (S$_{Zn}$)-[ZnL((S)-dpp)]). Conditions: rac-[Zn$_2$L$_2$] + (S)-dpp (2.50 equiv.); dry MS4A; C$_6$D$_6$; 4.00 mM; 70 °C; (1) 0 h, (2) 0.5 h, (3) 48 h. **b** Time-course variation of diastereomeric excess calculated from $^{19}$F NMR in various conditions. Conditions: rac-[Zn$_2$L$_2$] + (S)-dpp or (S)-mdp (2.50 or 1.00 equiv.), dry MS4A, C$_6$D$_6$, 4.00 mM, 70 °C. **c** The structure of (S)-mdp.

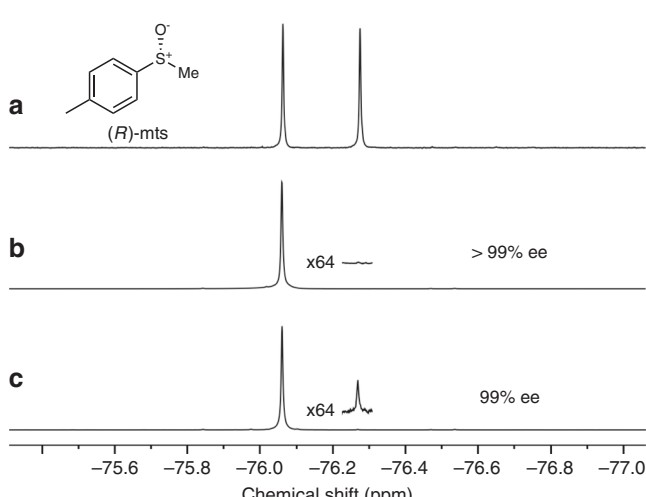

**Fig. 5 Evaluation of enantiopurity and stability of (S$_{Zn}$)-[ZnL(NC$^t$Bu)] by $^{19}$F NMR spectroscopy (C$_6$D$_6$, 300 K, 471 MHz). a** Structure of the chiral shift reagent (R)-mts and the reference spectrum with rac-[Zn$_2$L$_2$] + (R)-mts (1.10 equiv.). **b** Isolated (S$_{Zn}$)-[ZnL(NC$^t$Bu)] + (R)-mts (5 equiv.). **c** (S$_{Zn}$)-[ZnL(NC$^t$Bu)] was heated at 70 °C in C$_6$D$_6$ (8 mM) for one week, then (R)-mts (5 equiv.) was added to the reaction mixture. ee enantiomer excess.

As a potential candidate of the key intermediate of this catalysis, we synthesised aldehyde–catalyst complex [ZnL(2)], according to the typical mechanism of oxa-Diels–Alder reactions[34]. A racemate complex rac-[ZnL(2)] was successfully obtained from rac-[Zn$_2$L$_2$] and aldehyde 2. Its crystal and solution structures were determined by single-crystal X-ray

diffraction (Fig. 6b) and by $^1$H NMR spectroscopy including NOESY (Supplementary Fig. 2), respectively. The crystal structure shows that the oriented structure of the substrate 2 bound to the zinc centre is fixed by a CH–O hydrogen bond and a π–π interaction (Supplementary Fig. 3). In this crystal structure, the Si-face of aldehyde 2 in (R$_{Zn}$)-[ZnL(2)] (stereochemically corresponds to (S$_{Zn}$)-[ZnL(NC$^t$Bu)]) is blocked by the bulky mesityl group of the ligand (Fig. 6c). Although this complex was obtained under conditions different from those in the catalysis, the high enantioselectivity of the oxa-Diels–Alder reaction can be explained by a mechanism where diene 1 approaches from the Re-face of 2 in complex (R$_{Zn}$)-[ZnL(2)]. This Re-face approach is consistent with the absolute configuration of the product (R)-3, which was determined by single-crystal X-ray diffraction analysis of two derivatives (see Supplementary Methods).

In this scenario, the absolute configuration of the zinc centre during the reaction is maintained despite the catalytic cycle involving ligand exchange. To further support this scenario, $^{19}$F NMR measurements of the solution containing the chiral shift reagent (R)-mts and the reaction mixture showed that the enantiomer excess of the zinc catalyst after the reaction was > 99% ee (Supplementary Fig. 6). The labile coordination site of the complex (S$_{Zn}$)-[ZnL (NC$^t$Bu)] was therefore shown to be capable of effective and useful enantioselective catalysis.

In summary, a chiral-at-zinc complex has been efficiently synthesised with high enantioselectivity (> 99% ee) by utilising an achiral unsymmetric tridentate ligand and a chiral auxiliary ligand. The tridentate ligand imparts high stability to the zinc complex so that it can maintain enantiopurity as high as 99% ee even after heating at 70 °C for one week. Furthermore, the labile coordination site of this zinc complex has been found to function as the catalytically active centre of a highly enantioselective (87%

**Fig. 6 Asymmetric oxa-Diels–Alder reaction with an ($S_{Zn}$)-[ZnL(NC$^t$Bu)] catalyst. a** Reaction scheme. equiv. equivalents, ee enantiomer excess. **b** Single-crystal X-ray structure of *rac*-[ZnL(**2**)] (only the $R_{Zn}$ form is shown here) in a ball-and-stick model. Hydrogen atoms except in the formyl group are omitted for clarity. Colour code: Zn, blue grey; C, grey; N, blue; O, red; F, yellow green; S, yellow. Dashed magenta lines denote suggested noncovalent interactions (a hydrogen bond and a π–π interaction). **c** A schematic representation of a plausible origin of the enantioselectivity.

ee) asymmetric oxa-Diels–Alder reaction. This study demonstrates that Werner-type tetrahedral zinc centres can be utilised as a stable chirality source in the absence of chiral constituents. These findings expand the design concept for simple but versatile chiral metal complexes, and also are of great academic significance in overriding the common knowledge that tetrahedral zinc complexes are labile.

## Methods

**Chemicals.** Solvents and reagents were purchased from Tokyo Chemical Industry Co., Ltd., FUJIFILM Wako Pure Chemical Corporation, KANTO CHEMICAL CO.,INC. and Sigma-Aldrich Japan, and used without further purification unless otherwise noted. $C_6H_6$, $C_6D_6$, $CD_2Cl_2$, $^t$BuCN, hexamethyldisiloxane (HMDSO), 1,2-dichloroethane, tetrahydrofuran (THF) and $^i$PrOH were dehydrated over MS4A. $CH_3CN$ was dehydrated over MS3A.

**Handling.** All the manipulations regarding the zinc complexes were conducted under dry $N_2$ atmospheres using a UNICO UN-650F glove box connected with a Glovebox Japan GBJPWS3 gas purifier, a UNICO UL-1000A glove box connected with a UNICO MT-1000X gas purifier and gas-tight equipment, except for single-crystal X-ray diffraction and workup of catalysis.

**NMR.** $^1$H, $^{13}$C, $^{19}$F, $^{11}$B and 2D NMR spectra were recorded on a Bruker AVANCE III-500 (500 MHz) spectrometer. Tetramethylsilane (TMS) was used as an internal standard ($\delta$ 0 ppm) for the $^1$H and $^{13}$C NMR measurements when $CDCl_3$ was used as a solvent. A residual solvent signal was used for calibration of the $^1$H NMR measurement when $C_6D_6$ ($\delta$ 7.16 ppm), $CD_3CN$ ($\delta$ 1.94 ppm) or $CD_2Cl_2$ ($\delta$ 5.32 ppm) was used as a solvent. No corrections were conducted for $^{19}$F and $^{11}$B NMR measurement. Abbreviations: s, singlet; d, doublet; t, triplet; br, broad; m, multiplet. For the determination of the enantiomeric excess, high numbers of scans and data points were used.

**ESI-MS.** ESI-MS data were recorded on a Micromass LCT Premier XE ESI-TOF mass spectrometer. The experimental conditions were as follows: desolvation temperature, 150 °C; source temperature, 80 °C.

**Elemental analyses.** Elemental analyses (C, N, H) were conducted in the Micro-analytical Laboratory, Department of Chemistry, School of Science, the University of Tokyo using a Vario MICRO Cube elemental analyser with addition of MgO.

**Single-crystal X-ray diffraction.** Single-crystal X-ray diffraction analyses were performed using a Rigaku XtaLAB PRO MM007DW PILATUS diffractometer, and obtained data were processed by using CrysAlisPro 1.171.39.7e (Rigaku OD, 2015) software and analysed by Olex$^2$ 1.2.10 (OlexSys Ltd., 2018) software[35], using SHELXL-2017/1 software[36]. The crystals of the zinc complexes were handled in open air quickly before they were hydrolysed. For the details of single-crystal X-ray diffraction analyses of each compound, see the Supplementary Methods.

**CD.** CD spectra were recorded on a JASCO J-820 circular dichroism spectrometer. The experimental conditions were as follows: band width, 1 nm; response, 0.5 s; data acquisition intervals, 0.5 nm; scanning rate, 100 nm/min; number of scans, 4.

**HPLC.** HPLC data were collected on a JASCO MD-4010 photo diode array detector connected with PU-4185-Binary binary RHPLC semi-micro pump and CO-4060 column oven. The experimental conditions were as follows: slit width, 4 nm; data acquisition intervals, 4 nm; data integration width, 3 nm; sampling rate, 5 points/s.

The synthetic procedures, characterisation data and other reaction procedures are described in the Supplementary Methods.

## Data availability

Single-crystal X-ray diffraction data are available free of charge from the Cambridge Crystallographic Data Centre (https://www.ccdc.cam.ac.uk/data_request/cif) under reference numbers CCDC-1988762 (*rac*-[Zn$_2$L$_2$]), CCDC-1988761 (($S_{Zn}$)-[ZnL(($S$)-dpp)]), CCDC-1988763 (($S_{Zn}$)-[ZnL(NC$^t$Bu)]), CCDC-1988760 (*rac*-[ZnL(**2**)]), CCDC-2035788 (($R_{Zn}$)-[ZnL(($R$)-**3**)]) and CCDC-2033429 ((*R,R*)-**3′**). All other data generated or analysed during this study are available from the authors.

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

## Acknowledgements

We acknowledge Dr. Hiroyasu Sato (Rigaku Corporation) for refinement of the single-crystal X-ray diffraction data of (R,R)-3′. This research was supported by JSPS KAKENHI Grant No. JP16H06509 (Coordination Asymmetry) to M.S., Grant No. JP18K05138 to H.U. and Grant No. JP17J06579 (Grant-in-Aid for JSPS Fellows) to K.E.

## Author contributions

K.E. designed this work, performed experiments, analysed data and prepared the manuscript. Y.L. performed experiments and analysed data. H.U. designed this work, directed the research and prepared the manuscript. K.N. directed experiments. M.S. proposed and supervised this study and prepared the manuscript.

## Competing interests

The authors declare no competing interests.
