## [Peer Review File · Nature Communications]

REVIEWER COMMENTS

Reviewer #1 (Remarks to the Author):

The manuscript described the synthesis of a chiral-at-zinc complex, which was employed as a catalyst in an asymmetric oxa-Diels-Alder reaction. First an achiral, N-donating, tridentate, imino-amino-sulfonamide ligand was coordinated to Zn by reaction with ZnEt₂. The resulting dimeric, racemic-at metal complex was treated with a chiral auxiliary, (S)- α,α -diphenyl-2-pyrrolidinemethanol. Initially, a 1:1 diastereomeric mixture was obtained, which, after heating at 70 °C for 48 h, shifted to a 51:1 diastereomeric mixture of monomeric complexes, which differed only in the absolute configuration at the metal. The OH group of (S)- α,α -diphenyl-2-pyrrolidinemethanol assisted in the shift of the diastereomeric ratio from initially 1:1 to 51:5, as shown by comparison with (S)- α,α -diphenyl-2-pyrrolidinemethanol where the OH group was methylated. With the methylated chiral auxiliary, the diastereomeric shift slowed. The diastereomeric mixture was subsequently treated with pivalonitrile to replace the chiral auxiliary ligand. The enantiopurity of the resulting chiral-at-metal only complex was confirmed by ¹⁹F NMR and the chiral shift reagent methyl (R)-p-tolyl sulfoxide. After one week in C₆D₆ at 70 °C, virtually no racemization took place, demonstrating the configurational stability of the complex in aprotic solvents. The complex was finally tested as catalyst in an oxa-Diels-Alder reaction between Danishefsky's diene and 1-naphthaldehyde to obtain the dihydropyranone product with 87% ee (2 mol% catalyst load, 19 °C, 24 h). A transition state for the reaction is suggested based on an X-ray structure.

The research touches an important field. Chiral-at-metal complexes as enantioselective catalysts are far less investigated compared to chiral metal complexes, where the chiral information resides on the ligands. The study is for the most part well done. The compounds are well characterized. The chemistry described in the manuscript may inspire follow-up research. As such, I can recommend the manuscript for publication. However, the authors may consider the following points.

The authors describe the enantiomeric excess of the chiral-at-metal zinc complex as > 99.9% determined by ¹⁹F NMR with a chiral shift reagent. What is the detection limit of the method? The peak of the minor isomer is very small, and integration of the signal may not be very reliable. There is no doubt that the sample exhibits a high enantiomeric excess, but I am not certain if the accuracy of the method is that high.

The authors describe that the product of the catalysis experiment was "obtained quantitatively". It appears that the "yield" was determined by NMR using an (unidentified) internal standard. Is there a reason why the authors did not isolate the catalysis product? For the ee determination, the authors filtered the sample through a short pad of silica gel. Is it not possible to remove the solvent of the filtered sample to determine a weight of the product? As there is only one catalysis product, the authors should determine an isolated yield. Furthermore, the authors assign the absolute configuration of the major isomer of the catalysis "by the analogy to the case of PhCHO". While not necessarily unreasonable, it appears to me that the assignment of an absolute configuration based on "analogy" is uncommon. It is also not necessary for the present study, as the enantiomeric excess is the more important parameter. The authors may take the stereochemical assignment out of the manuscript.

The authors suggest a transition state based on an X-ray structure, where the oxygen of the aldehydes coordinates to the zinc. As the authors were able to obtain an X-ray structure, this assembly (i.e. the coordination of the aldehyde to the zinc center) must be fairly stable. However, "stable transition states" or intermediates actually slow down (or completely block) catalytic cycles. The existence of an X-ray structure alone is not sufficient proof that the assembly is an intermediate or transition state of the reaction. The authors should mention that other transition states are also possible.

In this context, the authors write "zinc centre is fixed by a CH–O hydrogen bond and a π – π interaction (Supplementary Fig. S6)". Figure S6 shows the CD spectra of the title compounds and I cannot see a relationship to the X-ray in Figure 4.

The authors write "thus preventing stereoinversion caused by partial dissociation or flattening (Supplementary Fig. S3)"; however, Figure S3 shows labelling of the protons of the zinc complexes and their characteristic NOEs. The authors write "That the tetrahedral structure is maintained in solution was supported by ^1H NMR measurements including NOESY in C_6D_6 (Supplementary Fig. S4)" but Figure 4 shows the process of the dynamic asymmetric induction. The authors write "were generated in a ratio of about 1:1, as demonstrated by ^1H and ^{19}F NMR spectroscopy (Supplementary Fig. S5)", but Figure S5 shows non-covalent interactions in the zinc complexes. The authors write "The Cotton effects of (SZn)-[ZnL(NCtBu)] were also confirmed in its CD spectrum (Supplementary Fig. S7)" but Figure S7 shows HPLC traces of the product of the catalytic reaction. The authors write "After acidic workup, adduct (R)-3 was obtained quantitatively with 87% ee (Supplementary Fig. S8)" but Figure S8 shows the configurational stability test. The authors write "the reaction mixture showed that the enantiomer excess of the zinc catalyst after the reaction was 99.8% ee (Supplementary Fig. S9)" but S9 shows the ligand synthesis.

In general, I find the manuscript a little difficult to read with that many references to Figures in the Supporting Information. The authors may consider taking some of these references to the supporting information out.

Reviewer #2 (Remarks to the Author):

This manuscript describes asymmetric construction of a tetrahedral chiral-at-zinc complex with high configurational stability using an unsymmetric tridentate ligand and one labile site was capable of performing highly selective asymmetric oxa-Diels-Alder reaction. It has long been a difficult task to construct and utilise Werner-type tetrahedral chiral metal centres without using chiral constituents because their stereoinversion is generally very fast, making it difficult to maintain long-term enantiopurity. A chiral-at-zinc complex was efficiently synthesised with extremely high enantioselectivity (> 99.9% ee) by utilising an achiral unsymmetric tridentate ligand, which imparted high stability to the zinc complex, and a chiral auxiliary ligand.

I am convinced that the results are important enough to be published in Nature Communications. Furthermore, there are some questions about the manuscript listed below:

1. Why choose zinc as the center metal? Have other metals been tried, like copper? If other metals do not work, what are the real advantages of zinc?
2. The unsymmetric tridentate ligand is very well designed, mainly using benzene as the rigid structure. Have other aromatic rings been tried, like naphthalene ring? Have the positions and steric hindrance of substituents on benzene rings been tried, except for isopropyl group?
3. There are some mistakes in the article file that need to be corrected, such as in the page 4, there are two places that describe as "Supplementary Fig. S2", which should be amended to "Supplementary Fig. S1". And the same error occurred on page 5 and page 6.
4. In the figure S7, there were impurity in HPLC spectra and the retention time of the racemates differed greatly from that of the catalytic system.
5. Some references should be cited, such as , a) Acc. Chem. Res. 2017, 50, 320–330 ; b) Acc. Chem. Res. 2017, 50, 2621-2431.

Reviewer #3 (Remarks to the Author):

I have reviewed the structure files submitted with the manuscript, "Asymmetric construction of

tetrahedral chiral zinc with high configurational stability and catalytic activity." I found the structures to be novel and the data and model refinement for those structures are excellent.

Reviewer #4 (Remarks to the Author):

The present manuscript describes the synthesis of chiral tetrahedral zinc complexes using restricted tridentate ligands.

The chiral octahedral and half-sandwich pseudotetrahedral complexes were already reported, however, the chiral tetrahedral metal complexes are rare. They investigated the configurational stability of chiral-at-zinc catalysts and catalytic activity carefully. Although the enantioselective reaction using chiral zinc complexes was only applied to Diels-Alder reaction of Danishefsky diene with naphthaldehyde and enantioselectivity was not excellent level, this example is one of a milestone for the synthesis and application for tetrahedral chiral metal complexes.

On the other hand, the key point for the synthesis of the chiral zinc complex is configurational stability of zinc center with (S)-dpp ligand. The manuscript described the hydroxy group in (S)-dpp accelerates the isomerization of the metal center, but it is not clear. Please provide the evidence of the effect for the hydroxy group. The result for the isomerization using (S)-dpp-like ligand without hydroxy group should be added in supporting information.

I can recommend the publication of the manuscript in Nature Communications with minor revision.

Responses to the reviewers' comments on NCOMMS-20-29229-T

We highly appreciate the reviewers' constructive comments on our manuscript for publication in *Nature Communications*. Below are our point-by-point responses to the reviewers' comments. To clarify the discussion, our response to each point is shown below alternately with the reviewer's comments. The reviewers' comments cited verbatim are italicized. The sentences added to the manuscript are highlighted with yellow backgrounds.

Responses to Reviewer 1:

The manuscript described the synthesis of a chiral-at-zinc complex, which was employed as a catalyst in an asymmetric oxa-Diels-Alder reaction. First an achiral, N-donating, tridentate, imino-amino-sulfonamide ligand was coordinated to Zn by reaction with ZnEt₂. The resulting dimeric, racemic-at metal complex was treated with a chiral auxiliary, (S)- α,α -diphenyl-2-pyrrolidinemethanol. Initially, a 1:1 diastereomeric mixture was obtained, which, after heating at 70 °C for 48 h, shifted to a 51:1 diastereomeric mixture of monomeric complexes, which differed only in the absolute configuration at the metal. The OH group of (S)- α,α -diphenyl-2-pyrrolidinemethanol assisted in the shift of the diastereomeric ratio from initially 1:1 to 51:5, as shown by comparison with (S)- α,α -diphenyl-2-pyrrolidinemethanol where the OH group was methylated. With the methylated chiral auxiliary, the diastereomeric shift slowed. The diastereomeric mixture was subsequently treated with pivalonitrile to replace the chiral auxiliary ligand. The enantiopurity of the resulting chiral-at-metal only complex was confirmed by ¹⁹F NMR and the chiral shift reagent methyl (R)-p-tolyl sulfoxide. After one week in C₆D₆ at 70 °C, virtually no racemization took place, demonstrating the configurational stability of the complex in aprotic solvents. The complex was finally tested as catalyst in an oxa-Diels-Alder reaction between Danishefsky's diene and 1-naphthaldehyde to obtain the dihydropyranone product with 87% ee (2 mol% catalyst load, 19 °C, 24 h). A transition state for the reaction is suggested based on an X-ray structure.

The research touches an important field. Chiral-at-metal complexes as enantioselective catalysts are far less investigated compared to chiral metal complexes, where the chiral information resides on the ligands. The study is for the most part well done. The compounds are well characterized. The chemistry described in the manuscript may inspire follow-up research. As such, I can recommend the manuscript for publication. However, the authors may consider the following points.

We highly appreciate the reviewer's profound understanding and professional commendation to our manuscript. We have sincerely considered the comments and suggestions as below.

The authors describe the enantiomeric excess of the chiral-at-metal zinc complex as > 99.9% determined by ^{19}F NMR with a chiral shift reagent. What is the detection limit of the method? The peak of the minor isomer is very small, and integration of the signal may not be very reliable. There is no doubt that the sample exhibits a high enantiomeric excess, but I am not certain if the accuracy of the method is that high.

The detection limit of this method can be determined by the signal-to-noise ratio of a peak height. IUPAC defines the detection limit to be generally 3.29 in a signal-to-noise ratio (ref. Currie, L. A. *Pure Appl. Chem.* **67**, 1699-1723 (1995)). In the spectrum in question (Fig. 4b), the signal-to-noise ratio of the major isomer signal was 21400, and in this measurement, peak heights are proportional to the concentration because two isomers have identical signal shapes (Fig. 4a). Therefore, the detection limit for the isomer ratio was calculated as $21400:3.29 = 6500:1$, which corresponds to 99.97% ee. In this way, high accuracy was supported here by considering the signal-to-noise ratios and by a spectrum with a high signal-to-noise ratio of the major isomer.

To clarify this point to readers, we add the following sentences to the Methods section of the revised manuscript on page 15:

“For the determination of the enantiomeric excess, high numbers of scans and data points were used. The detection limit reached 99.9% ee when the signal-to-noise ratio of the major isomer reached 6580, according to the IUPAC definition of detection limit $(3.29\sigma)^{35}$.”, and reference 35 is added (Currie,

L. A. Nomenclature in evaluation of analytical methods including detection and quantification capabilities (IUPAC recommendations 1995). *Pure Appl. Chem.* **67**, 1699-1723 (1995).

The authors describe that the product of the catalysis experiment was “obtained quantitatively”. It appears that the “yield” was determined by NMR using an (unidentified) internal standard. Is there a reason why the authors did not isolate the catalysis product? For the ee determination, the authors filtered the sample through a short pad of silica gel. Is it not possible to remove the solvent of the filtered sample to determine a weight of the product? As there is only one catalysis product, the authors should determine an isolated yield. Furthermore, the authors assign the absolute configuration of the major isomer of the catalysis “by the analogy to the case of PhCHO”. While not necessarily unreasonable, it appears to me that the assignment of an absolute configuration based on “analogy” is uncommon. It is also not necessary for the present study, as the enantiomeric excess is the more important parameter. The authors may take the stereochemical assignment out of the manuscript.

According to the reviewer’s suggestion, we conducted an additional experiment to isolate the catalysis product. We successfully isolated the product and obtained the isolated yield of 98%. We have revised the yield in the last paragraph on page 12 and Figure 6 and the experimental procedures to SI on page 15 and added a ¹H NMR spectrum of the isolated product to SI on page 40.

Furthermore, we conducted additional experiments to support the absolute configuration of the major enantiomer in the catalysis. We prepared two derivatives for single-crystal XRD (Figures shown below). Firstly, we conducted complexation of (*R*)-**3** and (*S*_{Zn})-[ZnL(NC^tBu)] to obtain (*R*_{Zn})-[ZnL((*R*)-**3**)]. Its single-crystal XRD analysis supported *R* configuration. Its correlation to the major enantiomer in the catalysis was further supported by HPLC analysis of (*R*)-**3** recovered from the crystals. Secondly, we chemically reduced the catalysis product. In this case as well, the single-crystal XRD result supported *R* configuration.

Thus, we could confirm the stereochemical assignment in the manuscript, and then added the following sentence in the first paragraph on page 14:

“This *Re*-face approach is consistent with the absolute configuration of the product (*R*)-3, which was determined by single crystal X-ray diffraction analysis of two derivatives (see Supplementary Information).”

We added “CCDC 2035788.cif”, “CCDC 2035788 checkcif.pdf”, “CCDC 2033429.cif” and “CCDC 2033429 checkcif.pdf” as additional supplementary files, which include the data of the crystal analysis. These data have been deposited in Cambridge Crystallographic Data Centre. We added acknowledgment to Dr. Hiroyasu Sato (Rigaku corporation) in the main text for his contribution to this single-crystal XRD analysis. The details of preparation, characterisation and the single-crystal XRD analyses of these derivatives are added to the revised SI on page 41.

Figure S1. Crystal structure of $(R_{Zn})\text{-[ZnL((R)-3)]}$. Ellipsoids are shown at 50% probability. Colour code: Zn, blue grey; C, grey; N, blue; O, red; F, yellow green; S, yellow.

Figure S58. Crystal structure of (R,R)-3'. Ellipsoids are shown at 50% probability. Colour code: C, grey; O, red.

The authors suggest a transition state based on an X-ray structure, where the oxygen of the aldehydes coordinates to the zinc. As the authors were able to obtain an X-ray structure, this assembly (i.e. the coordination of the aldehyde to the zinc center) must be fairly stable. However, “stable transition states” or intermediates actually slow down (or completely block) catalytic cycles. The existence of an X-ray structure alone is not sufficient proof that the assembly is an intermediate or transition state of the reaction. The authors should mention that other transition states are also possible.

We have a proper understanding that the X-ray structure alone is not sufficient proof of an intermediate structure of this catalysis. However, the reviewer’s argument “‘stable transition states’ or intermediates actually slow down (or completely block) catalytic cycles’ is not necessarily true. The reaction rate is not determined by the individual energy levels of transition states or intermediates, but it is determined by the activation energy. Even when the intermediate is stabilised, the reaction can be sped up if the transition state is more efficiently stabilised to decrease the activation energy.

Therefore, the stable aldehyde–catalyst adduct shown here can be a key intermediate in the catalytic cycle.

To further support this, we introduce a new reference about typical mechanisms of Lewis-acid-catalysed oxa-Diels–Alder reactions (Jorgensen, K. A. *Angew. Chem. Int. Ed.* **39**, 3558-3588 (2000)) (added as reference 34). In this reference, it is shown that catalytic reactions generally proceed via coordination of aldehydes to Lewis acids because the lowering of the aldehyde LUMO energy leads to a better interaction with the diene HOMO. Therefore, the aldehyde–catalyst adduct shown in the manuscript is highly likely to be a key intermediate.

To clarify these points to readers, we revised the beginning sentences of the first paragraph on page 13 to

“A key intermediate in this catalysis is most likely to be an aldehyde–catalyst complex [ZnL(2)], according to the typical mechanism of oxa-Diels–Alder reactions³⁴. A racemate of this complex *rac*-[ZnL(2)] was successfully obtained from *rac*-[Zn₂L₂] and aldehyde 2.” and reference 34 is added (Jorgensen, K. A. Catalytic asymmetric hetero-Diels-Alder reactions of carbonyl compounds and imines. *Angew. Chem. Int. Ed.* **39**, 3558-3588 (2000).) to the revised manuscript.

In this context, the authors write “zinc centre is fixed by a CH–O hydrogen bond and a π – π interaction (Supplementary Fig. S6)”. Figure S6 shows the CD spectra of the title compounds and I cannot see a relationship to the X-ray in Figure 4.

The authors write “thus preventing stereoinversion caused by partial dissociation or flattening (Supplementary Fig. S3)”; however, Figure S3 shows labelling of the protons of the zinc complexes and their characteristic NOEs. The authors write “That the tetrahedral structure is maintained in solution was supported by 1H NMR measurements including NOESY in C6D6 (Supplementary Fig. S4)” but Figure 4 shows the process of the dynamic asymmetric induction. The authors write “were generated in a ratio of about 1:1, as demonstrated by 1H and 19F NMR spectroscopy (Supplementary Fig. S5)”, but Figure S5 shows non-covalent interactions in the zinc complexes. The authors write “The Cotton effects of (SZn)-[ZnL(NCtBu)] were also confirmed in its CD spectrum (Supplementary

Fig. S7)” but Figure S7 shows HPLC traces of the product of the catalytic reaction. The authors write “After acidic workup, adduct (R)-3 was obtained quantitatively with 87% ee (Supplementary Fig. S8)” but Figure S8 shows the configurational stability test. The authors write “the reaction mixture showed that the enantiomer excess of the zinc catalyst after the reaction was 99.8% ee (Supplementary Fig. S9)” but S9 shows the ligand synthesis.

The reference numbers to the Supplementary Figures are shifted by 1 by error. We have corrected all of them in the revised manuscript.

In general, I find the manuscript a little difficult to read with that many references to Figures in the Supporting Information. The authors may consider taking some of these references to the supporting information out.

We have moved Supplementary Figures S1 and S4 to the main text as Figures 2 and 4 in the revised files.

Responses to Reviewer 2:

This manuscript describes asymmetric construction of a tetrahedral chiral-at-zinc complex with high configurational stability using an unsymmetric tridentate ligand and one labile site was capable of performing highly selective asymmetric oxa-Diels-Alder reaction. It has long been a difficult task to construct and utilise Werner-type tetrahedral chiral metal centres without using chiral constituents because their stereoinversion is generally very fast, making it difficult to maintain long-term enantiopurity. A chiral-at-zinc complex was efficiently synthesised with extremely high enantioselectivity (> 99.9% ee) by utilising an achiral unsymmetric tridentate ligand, which imparted high stability to the zinc complex, and a chiral auxiliary ligand. I am convinced that the results are important enough to be published in Nature Communications. Furthermore, there are some questions about the manuscript listed below:

The reviewer's thorough reading and valuable comments on our manuscript are highly appreciated. We have sincerely considered the comments and suggestions as below.

1. Why choose zinc as the center metal? Have other metals been tried, like copper? If other metals do not work, what are the real advantages of zinc?

We tried several kinds of metal ions at the initial stage of this research, such as Co^{2+} , Ni^{2+} , Cu^{2+} , Cd^{2+} , Al^{3+} and Ga^{3+} . Although some of them showed promising results, we focused on Zn^{2+} for two reasons. Firstly, tetrahedral Zn^{2+} is of particular interest because of its significance in biological systems. For example, there are many enzymes such as alcohol dehydrogenases which have a tetrahedral Zn^{2+} as a chiral centre as a catalytically active centre. This implies the potential usefulness of tetrahedral Zn^{2+} for catalysis and molecular recognition. This point is discussed in the introduction of our manuscript. Secondly, Zn^{2+} is one of the most challenging metal ions to conduct such a proof-of-concept research on tetrahedral chiral metal centres due to the coordinatively labile nature and the flexibility in the coordination numbers. Moreover, the diamagnetic nature with d^{10} electron configuration of Zn^{2+} enables in-depth NMR analysis, and also the redox inactivity prevents undesired redox reactions. We are now trying to use other transition labile metal ions for comparison, but this is beyond the focus of this work.

2. The unsymmetric tridentate ligand is very well designed, mainly using benzene as the rigid structure. Have other aromatic rings been tried, like naphthalene ring? Have the positions and steric hindrance of substituents on benzene rings been tried, except for isopropyl group?

So far, we have tried only benzene rings as rigid linkers and an isopropyl group as a substituent. We are now planning to change these moieties to other structures to investigate the correlation between ligand rigidity and configurational stability, but this is also beyond the focus of this work.

3. There are some mistakes in the article file that need to be corrected, such as in the page 4, there are two places that describe as “Supplementary Fig. S2”, which should be amended to “Supplementary Fig. S1”. And the same error occurred on page 5 and page 6.

Revised accordingly as this point was also pointed out by reviewer #1.

4. In the figure S7, there were impurity in HPLC spectra and the retention time of the racemates differed greatly from that of the catalytic system.

We conducted new experiments of the catalysis in a larger scale, isolated the product, and conducted HPLC analyses again. As a result, the impurities did not appear in HPLC chromatograms and the retention times of the racemate and the enantioenriched product corresponded well (see the figure below). We have replaced the HPLC chromatograms in Supplementary Fig. S5 and rewritten the experimental procedures on SI page 15.

Figure S2. HPLC profiles of **3**. (A) Racemic sample; (B) the sample obtained from the reaction with (S_{Zn}) -[ZnL(NC^tBu)].

5. Some references should be cited, such as , a) *Acc. Chem. Res.* 2017, 50, 320–330 ; b) *Acc. Chem. Res.* 2017, 50, 2621-2431.

We have added these two references to the revised manuscript as references 13 and 7.

Responses to Reviewer 3:

I have reviewed the structure files submitted with the manuscript, “Asymmetric construction of tetrahedral chiral zinc with high configurational stability and catalytic activity.” I found the structures to be novel and the data and model refinement for those structures are excellent.

We appreciate the reviewer’s professional examination of our data.

Responses to Reviewer 4:

The present manuscript describes the synthesis of chiral tetrahedral zinc complexes using restricted tridentate ligands.

The chiral octahedral and half-sandwich pseudotetrahedral complexes were already reported, however, the chiral tetrahedral metal complexes are rare. They investigated the configurational stability of chiral-at-zinc catalysts and catalytic activity carefully. Although the enantioselective reaction using chiral zinc complexes was only applied to Diels-Alder reaction of Danishefsky diene with naphthaldehyde and enantioselectivity was not excellent level, this example is one of a milestone for the synthesis and application for tetrahedral chiral metal complexes.

On the other hand, the key point for the synthesis of the chiral zinc complex is configurational stability of zinc center with (S)-dpp ligand. The manuscript described the hydroxy group in (S)-dpp accelerates the isomerization of the metal center, but it is not clear. Please provide the evidence of the effect for the hydroxy group. The result for the isomerization using (S)-dpp-like ligand without hydroxy group should be added in supporting information.

We highly appreciate the reviewer's valuable comments on our manuscript. A sentence "The result for the isomerization using (*S*)-dpp-like ligand without hydroxy group" is described in the original main text. To emphasise this point, we divided the first paragraph on page 8 into two parts, and modified the beginning part of the latter paragraph as follows: "To examine the effect of the hydroxy group of (*S*)-dpp on the stereoinversion, the isomerisation kinetics was compared with (*S*)-mdp (mdp = 2-(methoxydiphenylmethyl)pyrrolidine) where the hydroxyl group of (*S*)-dpp is methylated (Fig. 4c). As a result, It was found that (*S*)-dpp bearing a hydroxy group significantly accelerated the stereoinversion (Fig. 4b)."

In addition, Supplementary Fig. 4 has been moved to the main text as Fig. 4, according to the reviewer #1's comment.

I can recommend the publication of the manuscript in Nature Communications with minor revision.

REVIEWERS' COMMENTS

Reviewer #1 (Remarks to the Author):

The authors addressed many of the points I raised. I do not have any further comments at this point, except for some aspects I addressed in my original review.

With respect to the "X-ray structure of the intermediate", the authors at least write that a "key intermediate in this catalysis is most likely to be an aldehyde-catalyst complex". As I mentioned in my original review, I think such an intermediate is not necessarily unreasonable. However, its existence cannot unambiguously established by X-ray only (and the reference in support of the mechanism that the authors give does not explain the mechanism based on an X-ray structure). A stabilized intermediate may decrease the energy of activation, but it may also increase the energy for the intermediate to react further. At the end, the X-ray structure was not determined under "catalytic conditions", but without the diene substrate (which changes, of course, the chemistry). The authors could at least mention that an X-ray alone is no proof of an intermediate, as the X-ray was not determined under conditions that resemble the catalyzed reaction (i.e. the reaction between aldehyde and zinc complex was probably stoichiometric and in absence of the diene substrate).

I still disagree that the enantiomeric excess can be determined with the accuracy given by the authors. It is not clear to me how the NMR method relates to "3.29sigma". The literature cited by the authors does not specifically describe NMR methods, but is a general description of detection limits. If the author think their NMR method is that accurate, they needed to explain in detail, why so (not just by referring to theoretical considerations in the context of nomenclature). Whenever chiral shift reagents are used in the literature, the ee or er is given to the nearest number, like 99% (random examples: *Organic Letters* 2005, 7, 3985-3988; *Anal. Chem.* 2015, 87, 7258-7266; *Org. Lett.* 2006, 8, 5653-5655; *Inorg. Chem.* 2003, 42, 1378-1385).

In line 203, the authors write that "(R)-3 was obtained in 98% yield with 87.4% ee (Supplementary Fig. S5)". However, Figure S5 displays an ee of 94%.

Minor points. Add yield and ee of the catalysis product to the abstract. Caption Figure 6c "A plausible mechanism for enantioselective oxa-Diels-Alder reaction"; 6c does not show a mechanism but either a transition state or a graphical representation how the enantiomeric excess may materialize. The "Discussion" section is actually a summary or conclusion.

Reviewer #4 (Remarks to the Author):

I recently received this manuscript.

The revision points are revised, totally I can recommend the publication of the manuscript in *Nature Communications* without revision.

Responses to the reviewers' comments on NCOMMS-20-29229A

We highly appreciate the reviewers' continued cooperation and thoughtful reconsideration on our manuscript for publication in *Nature Communications*. Below are our point-by-point responses to the reviewers' comments. To clarify the discussion, our response to each point is shown below alternately with the reviewers' comments. The reviewers' comments cited verbatim are italicised and shown in blue. The sentences added to the manuscript are highlighted with yellow backgrounds.

Responses to Reviewer 1:

We profoundly appreciate the reviewer's re-examination of our manuscript. We have sincerely considered the comments and revised according to the suggestions as below.

The authors addressed many of the points I raised. I do not have any further comments at this point, except for some aspects I addressed in my original review.

With respect to the "X-ray structure of the intermediate", the authors at least write that a "key intermediate in this catalysis is most likely to be an aldehyde–catalyst complex". As I mentioned in my original review, I think such an intermediate is not necessarily unreasonable. However, its existence cannot unambiguously established by X-ray only (and the reference in support of the mechanism that the authors give does not explain the mechanism based on an X-ray structure). A stabilized intermediate may decrease the energy of activation, but it may also increase the energy for the intermediate to react further. At the end, the X-ray structure was not determined under "catalytic conditions", but without the diene substrate (which changes, of course, the chemistry). The authors could at least mention that an X-ray alone is no proof of an intermediate, as the X-ray was not determined under conditions that resemble the catalyzed reaction (i.e. the reaction between aldehyde and zinc complex was probably stoichiometric and in absence of the diene substrate).

According to the reviewer's suggestion, we have modified the first sentence on page 13 into "As a potential candidate of the key intermediate of this catalysis, we synthesised aldehyde–catalyst complex [ZnL(2)]..." and the first sentence on page 14 to "Although this complex was obtained

under conditions different from those in the catalysis, the high enantioselectivity of the oxa-Diels-Alder reaction can be explained by a mechanism...”.

I still disagree that the enantiomeric excess can be determined with the accuracy given by the authors. It is not clear to me how the NMR method relates to “3.29sigma”. The literature cited by the authors does not specifically describe NMR methods, but is a general description of detection limits. If the author think their NMR method is that accurate, they needed to explain in detail, why so (not just by referring to theoretical considerations in the context of nomenclature). Whenever chiral shift reagents are used in the literature, the ee or er is given to the nearest number, like 99% (random examples: Organic Letters 2005, 7, 3985-3988; Anal. Chem. 2015, 87, 7258–7266; Org. Lett. 2006, 8, 5653-5655; Inorg. Chem. 2003, 42, 1378-1385).

According to the reviewer’s suggestion, we have modified all the enantiomer excess values in the manuscript and the SI to two digits, removed description about the detection limits on page 15 and removed related Reference 35.

In line 203, the authors write that “(R)-3 was obtained in 98% yield with 87.4% ee (Supplementary Fig. S5)”. However, Figure S5 displays an ee of 94%.

We found that a wrong figure had been inserted as Supplementary Figure 5. We have replaced it with a correct figure showing 87% ee.

Minor points. Add yield and ee of the catalysis product to the abstract.

According to the reviewer’s suggestion, we have added the yield and ee of the catalysis product to the abstract.

Caption Figure 6c “A plausible mechanism for enantioselective oxa-Diels-Alder reaction”; 6c does not show a mechanism but either a transition state or a graphical representation how the enantiomeric excess may materialize.

According to the reviewer's suggestion, we have modified the legend of Figure 6c into "A schematic representation of a plausible origin of the enantioselectivity".

The "Discussion" section is actually a summary or conclusion.

According to the reviewer's suggestion, we have deleted the heading of "Discussion".

Responses to Reviewer 4:

I recently received this manuscript.

The revision points are revised, totally I can recommend the publication of the manuscript in Nature Communications without revision.

We highly appreciate the reviewer's thorough re-examination of our manuscript.